# Effects of Various Ripening Media on the Mesoporous Structure and Morphology of Hydroxyapatite Powders

**DOI:** 10.3390/nano13030418

**Published:** 2023-01-19

**Authors:** Margarita A. Goldberg, Olga S. Antonova, Nadezhda O. Donskaya, Alexander S. Fomin, Fadis F. Murzakhanov, Marat R. Gafurov, Anatoliy A. Konovalov, Artem A. Kotyakov, Alexander V. Leonov, Sergey V. Smirnov, Tatiana O. Obolkina, Egor A. Kudryavtsev, Sergey M. Barinov, Vladimir S. Komlev

**Affiliations:** 1A.A. Baikov Institute of Metallurgy and Materials Science, Russian Academy of Sciences, Moscow 119334, Russia; 2Institute of Physics, Kazan Federal University, 18 Kremlevskaya Str., Kazan 420008, Russia; 3Department of Chemistry, M.V. Lomonosov Moscow State University, Moscow 119991, Russia; 4Joint Research Center of Belgorod State National Research University «Technology and Materials», Belgorod State National Research University, Pobedy Str., 85, Belgorod 308015, Russia

**Keywords:** hydroxyapatite, aqueous precipitation, ripening in the mother solution, mesoporous powder, specific surface area, pore volume, length/width aspect ratio, zeta potential, electron paramagnetic resonance, non-Ostwald behavior

## Abstract

Mesoporous hydroxyapatite (HA) materials demonstrate advantages as catalysts and as support systems for catalysis, as adsorbent materials for removing contamination from soil and water, and as nanocarriers of functional agents for bone-related therapies. The present research demonstrates the possibility of the enlargement of the Brunauer–Emmett–Teller specific surface area (SSA), pore volume, and average pore diameter via changing the synthesis medium and ripening the material in the mother solution after the precipitation processes have been completed. HA powders were investigated via chemical analysis, X-ray diffraction analysis, Fourier-transform IR spectroscopy, transmission electron microscopy (TEM), and scanning (SEM) electron microscopy. Their SSA, pore volume, and pore-size distributions were determined via low-temperature nitrogen adsorption measurements, the zeta potential was established, and electron paramagnetic resonance (EPR) spectroscopy was performed. When the materials were synthesized in water–ethanol and water–acetone media, the SSA and total pore volume were 52.1 m^2^g^−1^ and 116.4 m^2^g^−1^, and 0.231 and 0.286 cm^3^g^−1^, respectively. After ripening for 21 days, the particle morphology changed, the length/width aspect ratio decreased, and looser and smaller powder agglomerates were obtained. These changes in their characteristics led to an increase in SSA for the water and water–ethanol samples, while pore volume demonstrated a multiplied increase for all samples, reaching 0.593 cm^3^g^−1^ for the water–acetone sample.

## 1. Introduction

Mesoporous materials characterized by the presence of porosity in the range of 2–50 nm, a sufficient surface area, and an essential pore volume are widely used in different industrial applications, particularly as catalysts, sorbents, sensors, and materials for optics and electronics, as well as in the field of medicine [1]. The most common mesoporous materials are based on zeolites and silica [2], including pillared materials [3], nanoporous carbon [4], and materials based on titanium [5,6], zirconium [7], cerium [8], as well as others [9]. Recently, hydroxyapatite (HA, Ca_10_(PO_4_)_6_(OH)_2_) powders started being considered as promising mesoporous inorganic compounds. Usually, HA plays a role as one of the most attractive calcium orthophosphates for biomedical applications in bone tissue restoration and drug delivery [10,11]. However, mesoporous HA materials are being applied to broadening application areas, and they are also suitable as catalysts and support systems for catalysis [12], and as adsorbents for the removal of contamination from soil [13,14] and water [15,16,17].

The creation of a low-cost and highly reproduceable synthesis technology for mesoporous HA is a challenge of today. Generally, the formation of the specific morphology and porosity could be achieved by the application of a templating system which would be removed during heating. The template could be organic, using Pluronic P123, Pluronic F127 [18], casein [19,20], or yeast cells [20], or inorganic, using CaCO_3_ [21], for example. Another way is linked with the application of organic compounds as synthesis modifiers. As such, polyvinylpyrrolidone (PVP) plays a role as the polymer modifier and provides the initial adsorption of Ca^2+^ with the further formation of HA crystals on the PVP’s surface. A further microwave treatment leads to the destruction of the PVP, resulting in the formation of pure mesoporous HA nanorods with a Brunauer–Emmett–Teller (BET) specific surface area (SSA) of 47 m^2^g^−1^ and a pore volume of 0.0097 cm^3^g^−1^ [22]. Chitosan is applied as an organic modifier during the precipitation synthesis, and HA with an SSA of 21–41 m^2^g^−1^ is exhibited [23]. When stearic acid is applied as an organic modifier for sol–gel synthesis and the pH level is increased to 11, the HA powders demonstrate an SSA of 66.2 m^2^g^−1^ with an average pore size of 5.86 nm. Additionally, the microemulsion [24], hydrothermal with template [25], and combustion [21] methods have been applied for HA synthesis. The introduction of cations has also led to HA structure distortion and significant SSA growth. In [22], the introduction of magnesium ions increased the SSA to 113.0 m^2^g^−1^, and the introduction of zinc ions led to an increase in the SSA to 136 m^2^g^−1^, as compared to 88 m^2^g^−1^ for pure HA [26]. In our previous papers, we demonstrated the influence of a procedure that includes Fe^3+^ doping during the precipitation synthesis and the subsequent ripening in the mother solution [27]. The Fe^3+^ doped powder with a theoretical formula of Ca_8.5_Fe_1_(PO_4_)_6_(OH)_2_ was characterized by an SSA of 194.5 m^2^g^−1^ when iron oxalate was used as a source of Fe^3+^ cations [27], while the ferrum chloride reactant contributed to the formation of materials with surface areas up to 141 m^2^g^−1^ [28]. Additionally, some efforts were addressed to creating a production route for HA from the waste source.

At the same time, ripening in the mother solution is one of the simple methods, which leads to an increase in the powder’s SSA. The growth of the SSA from 87.2 to 238.7 m^2^g^−1^ was achieved when composite HA-CaCO_3_ powders were ripened in a water-based mother solution for 21 days [29]. The investigations of the process of the CaCO_3_ powders ripening in the mother solution after the precipitation synthesis indicated the noticeable influence of organic liquids with reduced surface tensions, which were added to the water [30]. Thus, the SSA increased from 8–9 to 85 m^2^g^−1^ when ethanolamine was applied. Previously, the addition of an organic solvent influenced the formation of calcium-deficient HA with improved porosity during the hydrolysis of α-tricalcium phosphate under hydrothermal conditions [31]. Considering these approaches, in the present paper we investigated the influence of organic solvents on the HA powders’ characteristics after synthesis via the precipitation process and further ripening in the mother solution. We aimed to obtain mesoporous HA with improved textural properties, in particular SSA and pore volume, by changing the synthesis medium, and additionally via the ripening processes. Ethanol and acetone (dimethyl ketone) were chosen as models due to their complete solubility in water. The chemical and phase composition, cell parameters, functional groups, powder morphology, textural properties, and zeta potential were investigated, and electron paramagnetic resonance (EPR) spectra were recorded.

## 2. Materials and Methods

### 2.1. Sample Synthesis

The synthesis of the mesoporous HA powders with the theoretical formula Ca_10_(PO_4_)_6_OH_2_ was conducted using the following reactants: Calcium nitrate tetrahydrate (Ca(NO_3_)·4H_2_O, analytical grade, Ltd. “Labtech”, Moscow, Russia); diammonium phosphate (NH_4_)_2_HPO_4_, analytical grade, Ltd. “Himcity”, Moscow, Russia); 25% water solution of ammonia (NH_3_·H_2_O, extra-pure grade, Ltd. “Sigma-Tek”, Moscow, Russia); ethanol (C_2_H_6_O, 99%, Ltd. “Sigma-Tek”, Russia); and acetone ((CH_3_)_2_CO, 99%, Ltd. “Sigma-Tek”, Moscow, Russia). The syntheses were performed following reaction 1 via the precipitation method:10Ca(NO_3_)_2_ + 6(NH_4_)_2_HPO_4_ + 8NH_4_OH→Ca_10_(PO_4_)_6_OH_2_ + 20NH_4_NO_3_ + 6H_2_O(1)

For water–ethanol and water–acetone media synthesis, solutions of calcium nitrate and diammonium hydrogen phosphate were prepared by mixing deionized water with ethanol or acetone in a mass ratio of 1:1, respectively.

The diammonium hydrogen phosphate solution was added dropwise into the calcium nitrate solution during permanent mixing using an overhead glass-stirrer. The pH value of the reaction mixture was maintained at a level of 11–12 by adding aqueous ammonia. A gel pH meter (Testo, Lenzkirch, Germany) was used to control the pH level. After the end of the dropping process, the resulting suspensions were stirred for 2 h. Immediately after the stirring was concluded, the suspensions were divided into two equal parts before the sedimentation of the mother solutions and precipitates. The first part was decanted, washed with distilled water three times, filtered, and assigned the label “W-1” (water media), “WA-1” (water–acetone medium), or “WE-1” (water–ethanol medium). The second part of the suspension was ripened at 37 °C in a thermostat for 21 days and labeled “W-21”, “WA-21”, or “WE-21”, correspondingly. Before filtration, the mother solutions were decanted, and the precipitates were washed with distilled water three times. After filtration, the powders were dried at 60 °C for 24 h.

### 2.2. Sample Characterization

To investigate the chemical composition of the as-synthesized powders after 1 and 21 days of ripening, the powders were dissolved in a HCl-HNO_3_ mixture and analyzed via the high-precision technique of atomic emission spectrometry with inductively coupled plasma (AES-ICP, Vista Pro, Varian, Palo Alto, CA, USA). Wavelengths of 393,366 nm for calcium quantification and 214,914 for phosphorous quantification were applied.

The powdered materials were characterized via the X-ray diffraction (XRD) method (Shimadzu XRD-6000, Kyoto, Japan, CuK_α_ radiation, 10–60 (2θ)), with the identification of the phase composition conducted using the ICDD PDF2 database. Indexing of the peaks was carried out using card ICDD No. 09-0432 for HA and ICDD card No. 04-0777 for CaO. Quantitative phase analysis and lattice parameter estimation based on the Rietveld method were performed using PHAN% software [32].

Fourier-transform infrared (FTIR) absorption spectra of the samples were measured using the KBr method on a Nicolet Avatar 330 FTIR spectrometer (ThermoFisher Scientific, Waltham, MA, USA), and spectra were obtained in the range from 4000 to 400 cm^–1^ so as to evaluate the functional groups of the specimens. 

The particle morphology investigations were conducted via transmission electron microscopy (TEM, JEOL JEM 2100, Tokyo, Japan) on the carbon-sputtered specimens, with an accelerating voltage of 200 kV, 1000 mm of the diffraction length, 25.1. The calculation of the particle size was conducted using ImageJ software by measuring 50 random particles on each of the images, followed by statistical analysis. Identification and indexing of the selected-area electron diffractions (SAED) were performed with CrysTBox software [33]. Scanning electron microscopy (SEM) was performed on a Tescan Vega II microscope (Tescan, Brno, Czech Republic), and the powders were investigated without any sputtering.

The specific surface area (SSA) was determined according to Brunauer, Emmet, and Teller (BET), and pore-volume and pore-size distributions of the as-synthesized powders were determined using the Barret, Joyner, and Halenda analyzer model (BJH) using the low-temperature nitrogen adsorption measurements (Micromeritics TriStar Analyzer, Micromeritics Instrument Corporation, Norcross, GA, USA).

The zeta potential was measured for the standard suspensions of particles in distilled water, which were prepared at a concentration of 5 µg/10 mL using Photocor Compact—Z equipment (Tallin, Estonia). The suspensions were subjected to dispersion in an ultrasonic bath for 30 min to remove agglomerates and ensure the uniform distribution of particles in the water. Then, 500 μL of the suspension was placed in a measuring cuvette, and the total volume was adjusted to 2 mL with distilled water.

Electron paramagnetic resonance (EPR) measurements were performed in the pulsed X-band (ν_mw_ = 9.6 GHz) modes on a Bruker Elexsys E580 spectrometer (Billerica, MA, USA). Field-swept electron spin-echo (ESE) spectra were recorded with the standard pulse sequence π/2 − τ − π with a π/2 pulse duration of 16 ns for X-band, and a minimal time delay τ = 240 ns. T_2e_ was studied by tracking the primary ESE amplitude with the same π/2 − π pulse durations while varying τ with a minimal possible step of 4 ns. T_1e_ was extracted from an inversion recovery experiment by applying the π − T_delay_ − π/2 − τ − π pulse sequence, where both the π pulse duration and τ are fixed while T_delay_ is varied. The electron–nuclear interactions were analyzed using a three-pulse electron spin-echo envelope modulation (ESEEM) sequence (π/2 − π/2 − π/2—ESE), with a change in both distances (τ and T) from 240 ns to 1204 ns. The collected ESEEM results in the time domain were further analyzed using Origin Pro 2017 software by subtracting the exponential curve to gain nuclear modulations and the following Fourier transform. All EPR and ESEEM measurements were conducted at room temperature (T = 297 K).

## 3. Results

### 3.1. Chemical Compositions of the Powders

The results of the chemical composition investigations demonstrated the formation of calcium-enriched HA materials (Table 1). Ripening in the mother solution resulted in the further growth of the Ca/P ratio, indicating the continuous processes of the HA crystal’s interaction with the mother solution with Ca^2+^ incorporation in the structure. The HPO_4_^2−^ groups will be presented below for the as-synthesized powders according to FTIR data, indicating the Ca-deficient HA formation. The HA could be formed in the Ca/P ratio of 1.67 to 2.0 [34] with a noticeable difference from the theoretical ratio of HA (1.67). The observed data could be linked with the formation of carbonated-HA with a replacement of PO_4_^3−^ on the CO_3_^2−^ groups.

### 3.2. Phase Compositions of the Powders

According to the XRD analyses, all of the powders were characterized by only the apatite phase (ICDD card No: 9-432). After the synthesis, the degree of crystallinity was the highest in the case of the water medium synthesis (W-1) samples, and it decreased when the organic solvents were used (Figure 1a). After ripening, the degree of crystallinity rose, and additional peaks at 32.7–32.9 2Θ were evident. The calculation of the cell parameters demonstrated that the a parameter decreased during ripening regardless of the medium, while the c parameter grew insignificantly in the water (W-21) and water–acetone (WA-21) media (Table 2). On the contrary, the water–ethanol (WE-21) samples demonstrated significant growth in the c parameter, from 0.6870(2) nm to 0.6882(5) nm, after ripening. As a result, the c/a ratio and the HA structure became closer to the theoretical one, contributing to the changes in the powder morphology.

The SAED data confirmed the results of the XRD analyses, indicating the formation of the pure HA phase for all samples. The SAED ring identification was performed using CrysTBox software [33]. The degree of crystallinity dropped from the W-1 to the WE-1 samples, but ripening in the mother solution for 21 days resulted in the appearance of single-dot reflexes on the powder ring diffractograms, indicating a growth in crystallinity in all of the ripening media. The estimated Miller indexes are presented in Figure 2.

### 3.3. FTIR Investigation

The FTIR spectra are presented in Figure 3, and they confirmed the formation of the HA structure [35]. However, the spectra of the 1-day samples seem smoother and more poorly resolved compared to the 21-day spectra. The resolution of the hydroxyl group OH^−^ bands at the 3571 cm^−1^ and 633 cm^−1^ of the powders, obtained after 1 day of ripening, was noticeably affected by the synthesis medium. The lowest spectral intensity was observed for the WA-1 sample; the band at 633 cm^−1^ was not observed at all. In the case of the 21-day samples, the bands at 3571 cm^−1^ of hydroxyl were well-resolved for all samples, with the highest intensity for the W-21 sample and lowest for the WE-21 sample. The ν_2_ (H-O-H) was evident at 1635 cm^−1^. Additionally, the phosphate groups decreased in their intensity when the ethanol- and acetone-containing media were applied. The asymmetric stretching vibration mode ν_3_ of PO_4_^3−^ was present at 1094 and 1081 cm^−1^; the ν_1_ PO_4_^3−^ symmetric stretching vibration mode was observed at 960 cm^−1^; the ν_4_ O-P-O bend was characterized by the bands at 600 and 563 cm^−1^; and the double-generated bending mode (ν_2_) PO_4_^3−^ was observed at 472 cm^−1^ [36]. A broad peak ranging from 3700 to 2500 cm^−1^ could be attributed to adsorbed water, and the maximum at 3135 cm^−1^ could be associated with adsorbed NH_3_ as a synthesis co-product [37]. Bands of the adsorbed CO_2_ were observed at 2424, 2393, 2354, and 2327 cm^−1^. The bands attributed to residual nitrates were present in all samples at 1382–1387 cm^−1^, (ν_1_ + ν_4_) NO^3^ at 1760 cm^−1^. For the W-1 and WA-1 spectra, an additional nitrate band appeared at 825 cm^−1^.

The CO_2_ may not only adsorb on the surface of the sample but may also introduce the HA structure, forming A, B, or AB types of carbonated hydroxyapatite, respectively [35]. Thus, the samples ripened for 1 day were characterized by bands at 1401, 1455, and 873 cm^−1^, which denoted that CO_2_ predominantly formed a B-type CO_3_^2−^ substitution [38]. Additional weak bands at 1542 cm^−1^ and a broadening of the 873 cm^−1^ band reflected the minor presence of an A-type CO_2_ substitution for WA-1. For all samples, after ripening for 21 days, we noted the broadening of the band positioned at 875 cm^−1^, which could be simultaneously related to A-type (878 cm^−1^) and B-type (873 cm^−1^) carbonates, as well as to hydrophosphate (HPO_4_^2−^). These could indicate the formation of an AB-type carbonate substitution. For W-21, bands at 1419, 1455, and 1550 cm^−1^ confirmed an AB-type substitution, while for WE-21 and especially for WA-21, the disappearance of the band at 1550 cm^−1^ ν_3_(CO_3_^2−^) indicates the predominating formation of B-type substituted HA and may reflect the contribution of HPO_4_^2−^ in the 875 cm^−1^ band’s broadening.

Thus, according to the FTIR data, the 21-day ripening process led to a significant increase in the overall spectral resolution, particularly for the OH^−^ bands at 3571 and 633 cm^−1^. The spectra of W-21 were characterized by the highest intensity of these bands; the WE-21 and WA-21 spectra demonstrated the more pronounceable difference between the ripened and the as-synthesized materials. The broad peaks of the adsorbed water, CO_2_, and NH_3_ became weaker, indicating a decrease in the adsorption capacity of the materials. The carbonate introduction in the HA structure seemed to predominantly lead to the formation of B-type substituted HA.

### 3.4. Transmission and Scanning Microscopy Investigations

According to the TEM data, the morphology of the W-1 samples was characterized by the formation of ellipsoids, rods, and several particles that were close to a spherical shape, with an average length (L) of 36 with a standard deviation (SD) of ± 11 nm, and an average width (W) of 14 ± 4 nm (SD); the aspect ratio (L/W) was 2.57 (Figure 4 and Appendix A). The introduction of organic solvents into the synthesis media resulted in significant growth in the aspect ratio (L/W). Thus, in WA-1, we observed the formation of needles with a L = 28 ± 8 nm, a W = 5 ± 2 nm, and an L/W aspect ratio = 5.6. The WE-1 materials presented with elongated needles with an L/W aspect ratio of 8.33, a L = 50 ± 21, and a W = 6 ± 2 nm. Goto et al. previously demonstrated an increase in the length and a decrease in the width, occurring with an increase in the ethanol fraction, in the formation of the needle-like crystals, with an L up to 4.5 µm when HA was synthesized in the water–ethanol medium under solvothermal conditions [39]. In the present experiment, a similar pronounceable anisotropy of the crystals was observed, and this confirmed the XRD cell-parameter calculation results.

Ripening in the mother solution for 21 days resulted in noticeable changes in the morphology of the particles. Thus, the aspect ratio for all the samples became lower, indicating the formation of more equilibrium shapes (Appendix A). Particles were characterized by the following values: for W-21, they were: L = 30 ± 11 nm, W = 18 ± 6 nm, L/W aspect ratio = 1.67 (rods and close-to-spherical particles); for WA-21, they were: L = 25 ± 12 nm, W = 15 ± 5 nm, L/W aspect ratio = 1.67 (rods, spherical particles, and several needles); for WE-21, they were: L = 56 ± 38 nm, W = 14 ± 7 nm, L/W aspect ratio = 4.00 (particles were predominantly of a heterogeneous morphology, including needles, spheres, and rods, in apparently equal quantities).

Thus, the appearance of the new, small particles and the formation of shorter, but wider, crystals, which lie more separately as shown in the figures, were observed after the ripening process.

The particles of all the samples formed agglomerates, and the spaces between them contributed to the porosity of the material. According to the SEM (Figure 5) and TEM results, the process of ripening in the mother solution contributed to the formation of the looser agglomerates, as compared to the samples after synthesis. The decrease in the degree of powder agglomeration was further confirmed by measuring the zeta potential.

### 3.5. Textural Properties of the Powders According to Measures of Nitrogen Adsorption

According to the BET data, the powders’ adsorption–desorption isotherms were characterized as type-IV according to the IUPAC as evident in the formation of the mesoporous materials (Figure 6) [40]. The hysteresis loops of the W-1 and WE-1 samples were attributed to H3-type behavior. This type is characterized by split-shaped pores caused by the aggregation of plane-parallel and rod-like particles [28]. The powders obtained via the water–acetone medium (WA-1) were characterized by the H2-type hysteresis loop, which is typical for a corpuscular structure with an irregular pore shape. After ripening, all of the powders were characterized by the presence of an H3-type hysteresis loop and micropores according to the t-plot.

When the materials were synthesized in the water–ethanol and water–acetone media, the growth of the SSA and total pore volume compared to those synthesized in the water medium was substantial: from 30.3 m^2^g^−1^ to 52.1 m^2^g^−1^ and 116.4 m^2^g^−1^, and from 0.08 cm^3^g^−1^ to 0.231 and 0.286 cm^3^g^−1^, respectively (Table 3). The achieved SSA and pore-volume values are comparable to the ones that have been demonstrated for mesoporous silica materials [41]. The observed data are in good agreement with the TEM results, which indicated that the WA-1 particles were characterized by a smaller size, as compared to the powders synthesized in the water and water–ethanol media. At the same time, the WE-1 samples were characterized by a noticeable morphological anisotropies, which contributed to the porosity growth.

After ripening in the mother solution for 21 days, the most significant increase in the SSA value, that being more than 2 times the amount, was achieved for the W-21 sample—up to 79.4 m^2^g^−1^. This tendency towards a growth in the SSA was observed for WE-21, which reached a value of 83.2 m^2^g^−1^. The SSA value of WA-1 and WA-21 changed insignificantly, with a slight decrease after ripening. Simultaneously, the powders demonstrated a growth in total pore volume of more than 2 times for the water–acetone (up to 0.593 cm^3^g^−1^) and the water–ethanol (up to 0.486 cm^3^g^−1^) ripening media, and a growth of 4.4 times for the water-based ripening medium, up to 0.353 cm^3^g^−1^, exceeding all values obtained for the as-synthesized powders. These values were higher than one presented early for pore-expanding mesoporous HA (0.36 cm^3^ g^−1^), obtained by the 1,3,5-trimethyl benzene and cetyltrimethylammonium bromide template method [42]. In addition, the micropore volume increased for the WA-21 and WE-21 samples according to the t-plot data. The pore-size distributions, as computed based on the BJH analysis, indicated pore-size expansion after ripening, characterized by enlarged adsorption, and an average pore width in the range of 17.8–21.3 nm.

### 3.6. Zeta Potential

The zeta potential measurements demonstrated a decrease in the potential values after the ripening process (Figure 7) [43,44]. The similarly strongly negative values of the HA powder suspensions were previously demonstrated by Li D. et al. [43] and Zhang N. et al. [44] The zeta potential values of the WE samples were the lowest, reaching −14.17 ± 1.04 mV and −18.11 ± 0.82 mV at a pH value of 7 for WE-1 and WE-21, respectively. It is important to note that materials synthesized in all three media demonstrated insignificant differences before ripening, and these grew noticeably after ripening, which contributed to the materials’ crystallization. These could be linked with a change in the particles’ morphology, as well as to the number of CO_3_^2−^ and HPO_4_^2−^ groups introduced into the HA structure.

### 3.7. EPR Spectroscopy

The structure of pure HA does not contain paramagnetic atoms (with a non-zero magnetic moment), so there was no resonant absorption signal. The procedure of the chemical precipitation synthesis of the samples was accompanied by the incorporation of nitrate anions into the positions of phosphorus groups ${PO}_{4}$. Because of the difference between the number of oxygen bonds, for the nitrate-containing hydroxyapatites, there was a negatively charged anion ${NO}_{3}^{-}$, which served as an electron trap. X-ray irradiation of the samples led to the formation of free electrons captured by nitrate anions, followed by the formation of the nitrogen radical ${NO}_{3}^{-}$ with electron spin S = ½ [45].

The recorded EPR spectra (in pulse mode at room temperature) are shown in Figure 8. The mentioned complex—nitrogen radical, due to the presence of a nuclear magnetic moment in nitrogen ^14^N, can serve as a spin label or “probe” for the analysis of the local environment, providing information about its structural changes under various effects on the sample [46]. This fact is also confirmed by the electron spin-echo envelop modulation (ESEEM) spectrum, which contains resonant signals from hydrogen and phosphorus nuclei (Figure 9). The resonance lines in the frequency domain are located following the Larmor frequencies of the nuclei, depending on the fixed magnetic field (B_0_) and the nuclear gyromagnetic ratio (γ_n_): ν_Larmor_ = γ_n_*B_0_. This result indicates that the nitrogen radical is contained in the crystal lattice of the sample and surrounded by structural elements (^1^H and ^31^P). The presence of three anisotropic components in the EPR spectra (Figure 8) is caused by the hyperfine dipole–dipole interaction between the electronic and nuclear magnetic moments. As can be seen in Figure 8a,b, the comparative analysis shows no difference between the EPR results obtained from the samples synthesized in the different media (water, ethylene, and acetone). The results, depending on the synthesis medium, are the same for each ripening period (1 day (as-synthesized) or 21 days). Obvious differences in the spectra are observed for the samples, depending on the ripening periods of 1 and 21 days (Figure 8c). The simulation and the description of the EPR spectra using the spin Hamiltonian (2), which includes the anisotropic Zeeman term and the hyperfine interaction tensor (presented in Figure 5c as a green, dashed line), are obtained as follows:(2)$$Ĥ=g_{\left|\right|}\beta B_{z}{Ŝ}_{z}+g_{⊥}\beta \left(B_{x}{Ŝ}_{x}+B_{y}{Ŝ}_{y}\right)+A_{\left|\right|}{Ŝ}_{z}{Î}_{z}+A_{⊥}\left({Ŝ}_{x}{Î}_{x}+{Ŝ}_{y}{Î}_{y}\right)$$
where g is the spectroscopic splitting factor, β is the boron magneton, B is the magnetic field projection, A is the magnitude of the hyperfine interaction, and S and I are the projections of the electron and nuclear magnetic moments, respectively. Calculated spin-Hamiltonian values are shown in Table 4.

Based on the EPR results, it can be argued that the magnitude of the hyperfine interaction A decreases after 21 days of ripening, and the EPR lines become narrower and more resolved (see Table 4). A similar effect is due to the fact that, during ripening, nanoparticle material becomes more crystallized, with fewer defects (dislocations) in the crystal lattice, as confirmed by XRD results. It is worth noting that both components of hyperfine interaction A_||_ = a + 2b and A_⊥_ = a − b, where a is the isotropic contact Fermi interaction, and b is the anisotropic dipole–dipole interaction, decreased by approximately the same amount, which indicates that the electron density distribution becomes more localized for the nitrogen radical [47]. The EPR spectra for samples after the synthesis contain an additional signal (g_⊥_ = 2.005 and g_||_ = 2.000), which disappears after 21 days of ripening. The nature of this center may be associated with the presence of broken bonds or with the presence of a carbonate radical which was displaced with ripening.

The spin–spin T_2_ = 2.9 ± 4 µs and spin–lattice T_1_ = 28 ± 3 µs relaxation times measured for the irradiated hydroxyapatite samples did not depend on the type of medium or the ripening time, which indicates the initial homogeneous distribution of nitrate radicals within the sample, the absence of side phases of synthesis (octacalcium phosphate, tricalcium phosphate, etc.), and the retention of their initial unit cell parameters.

## 4. Discussion

The simple and low-cost synthesis of micro- and mesoporous HA powders is a challenge for today [48]. The fields for its application are broadening every day, especially in catalysis [49,50,51,52,53]; in the removal of environmental pollution in the soil, wastewater [15,54], and air [55,56,57]; and in the biomedical [58,59,60] area. Additionally, mesoporous HA is considered a potential material for optical applications [61], as an MRI contrast agent [62], in chromatography matrices [63], and as a carrier for the food industry [64]. The precipitation route is characterized by simple processing, a low cost, and ease of application in industrial production [65]. In our previous works, we demonstrated the effect of ripening HA in a mother solution when distilled water was applied as a synthesis medium for HA-CaCO_3_ materials [29], and we presented opportunities for increasing HA SSA and pore volume based on doping with different cations [28,66]. The influence of introducing ethanol, acetone, and ethanolamine to the water synthesis medium and the process of further ripening in mother solutions was discussed in [30] for CaCO_3_ powders, but the SSA increased noticeably only in the case of the water–ethanolamine mixture application [30] In the present study, the influences of the synthesis and the ripening medium on the properties of un-doped, pure HA powders were established.

First, we demonstrated the strong influence of the addition of acetone to the synthesis medium on the textural properties of the as-synthesized HA powders. Previously, the increase of the SSA was demonstrated for the modified precipitation method with the formation of nanoemulsions when Ca(NO_3_)_2_ × 4H_2_O was dissolved in pure acetone and further mixed with an aqueous solution of (NH_4_)_2_HPO_4_ and NH_4_HCO_3_. We obtained carbonated HA with an SSA of 43.67 m^2^g^−1^ [67]. Additionally, increases in the SSA from 66.7 to 75.2 m^2^g^−1^ and the pore volume from 0.119 to 0.122 cm^3^g^−1^ were demonstrated for the mesoporous esterified HA powders with 2-bromo-2-methylpropionic, when 30% of the acetone solution was added to the water solution to prevent the powders’ agglomeration [68]. In the present work, we obtained WA-1 mesoporous powders with an SSA of 116.4 m^2^g^−1^ and a pore volume of 0.286 cm^3^g^−1^ due to the mixing of the both water–acetone solution of Ca(NO_3_)_2_ × 4H_2_O and (NH_4_)_2_HPO_4_. These unexpectedly high textural properties could be comparable with the values achieved by multistage, template-based synthesis, such as in the case of casein or Pluronic surfactants [19,69]. The ripening in the mother solution of the WA samples resulted in the increase of the degree of crystallinity according to the SAED data and the XRD results, as well as the FTIR spectral resolution. At the same time, a slight decrease in the SSA (about 6%) was observed, but the porosity doubled and reached 0.593 cm^3^g^−1^ (for WA-21). Similar, multiple increases in the pore volume after ripening were observed for the powders obtained via the water-based (W-21) and water–ethanol-based (WE-21) synthesis. Ripening in the mother solution resulted in the enlargement of the spaces between particles, with a corresponding growth in the adsorption average pore width. This behavior was associated with significant changes in the morphology, as confirmed by the drop in the aspect ratio according to the TEM results analysis. The TEM and SEM data also reflected a decrease in the density of the material’s agglomeration, and segregated particles were observed on the microphotographs (Figure 4 and Figure 5). We also established the modulus growth in zeta-potential values (the negative charge decreased) after ripening in the water–ethanol and water–acetone media, and we evaluated the contribution of the double-electron layers to the particles’ separation from each other, with the resulting growth in the pore volume.

Ripening in the mother solution is expected to occur in accordance with the Ostwald ripening of the nanoparticles. The Ostwald ripening is linked to the growth of the particle size due to the recrystallization of the smaller crystals on the surface of the large ones. According to the obtained results, we observed the change of the aspect ratio with a decrease in the L and an increase in the W of the crystals. Thus, the non-Ostwaldian behavior of HA particles was established, as it was previously for Bi_2_WO_6_ [70] and for NaCl crystals [71].

The differences in the behaviors of the powders during the synthesis and ripening processes could be linked to the difference in the dielectric constants, which decrease from water to acetone as ε(W) = 80.20, ε(E) = 25.16 ε(A) = 21.13 [72,73], and correspondingly, the constant values of the fluid’s mixture drop. The influence of the dielectric constants during the synthesis from alpha-tricalcium phosphate (α-TCP) at the solvothermal conditions in the ethanol–water medium was suggested as a reason for the different HA powders’ morphological formations [31,39]. The processes of the solvothermal synthesis are linked with the simultaneous dissolution of the initial α-TCP and its recrystallization in the presence of the OH^−^ anions. With the decreasing dielectric constant of the solvent, the solubility decreases due to the drop in the solvation energy, and the solution becomes supersaturated with Ca^2+^ and PO_4_^3−^ ions with the formation of small crystals [74,75]. In the case of the precipitation method, when ethanol and acetone were added to the initial solutions, the synthesis mixture became supersaturated more rapidly, and the nucleation process became dominant over the crystal growth. Thus, as acetone has the lowest dielectric constants, the initial, smaller particles were formed during the interaction of the dissolved Ca^2+^, PO_4_^3−^, and OH^−^, which was reflected in the TEM and BET data. The water–ethanol medium induced the formation of whisker-like particles, as in the case of the solvothermal synthesis. Ripening in the mother solution resulted in recrystallization, with the broadening of the whiskers and the formation of additional plate-like crystals.

Previously, the influence of the aspect ratio on the surface area was investigated by the measurement of barreling contour changing in a model of a cylindrical aluminum specimen which were treated by the upsetting machine [76]. It was established that the total surface area first decreases at low upset ratios for long specimens, and then it begins to increase for middle-length and short specimens; the total surface area increases as the upsetting ratio rises. This barreling phenomenon was linked to the decreased contribution of the lateral surface in the total surface when length/height declined and was presented by a mathematical solution. Thus, the increase in the total measured geometric surface area was established when samples became shorter and wider with round lateral surfaces. Thus, we could expect a similar tendency of the changing of the lateral surface contribution in the particles’ SSAs when the aspect ratio decreases during the ripening process due to morphological change. In addition, the increase in the SSA could be linked to the more-isolated particle stage during the formation of the smaller and looser agglomerations. These led to an increase in the nitrogen adsorption capacity on the particles’ hole surfaces. We could assume that an increase in the crystallinity of the powders (as confirmed by the XRD, SAED, and EPR data) could provide the particles with a more consummate surface and lower charges, which influence the repulsion of particles from each other. As the water–acetone powders were characterized by the most varied morphology, the denser packing of the particles could be realized, and the SSA value did not grow during ripening [77]. Additionally, we could not reach the geometric factor of the surface growth as particles were changed to the lower, minimal point. (The upset ratio of 20% was a minimal point in the case of the metal specimens) [76]. At the same time, the tendency of the aspect ratio to decrease was preserved, and pore volume increased.

Thus, the synthesis and ripening media significantly influence the morphology and textural properties of HA powders. As the interest in HA as a promising catalyst for a variety of catalytic reactions has increased exponentially over the past 3 decades [78], the modification of its mesoporous structure could provide significant benefits in the green chemistry industry, making it a worthy competitor to silica and alumina catalysts [79].

## 5. Conclusions

In the present paper, we presented the possibility of increasing the SSA and pore volume of HA powders based on changing the synthesis medium—from pure water to a water–ethanol and a water–acetone mixture. When the water–acetone mixture was applied, the formation of HA powders with an SSA value of 116.4 m^2^·g^−1^ was achieved. Additionally, we established, that HA particles crystallize during a ripening period in the mother solution, but the process is characterized by non-Ostwald behavior. We observed the phenomenon that, instead of the small particles dissolving, the powders recrystallized during the ripening process with a corresponding decrease in their length/width aspect ratio. These led to the formation of more equiaxed and equilibrium particles, which was confirmed by the XRD, FTIR, and EPR data. At the same time, changes in the zeta-potential values and the formation of smaller, looser agglomerates were observed. This led to an increase in the SSA values for the W-21 and WE-21 samples and multiplied pore-volume growth for all of the powders. The developed synthesis route is low-cost and uncomplicated, but it provides the possibility to obtain powders with textural properties comparable to mesoporous materials synthesized using special templates and labor-intensive procedures. The mesoporous HA powders developed herein are promising for industrial applications in the areas of catalysis, biomedicine, and sorption.

## Figures and Tables

**Figure 1 nanomaterials-13-00418-f001:**
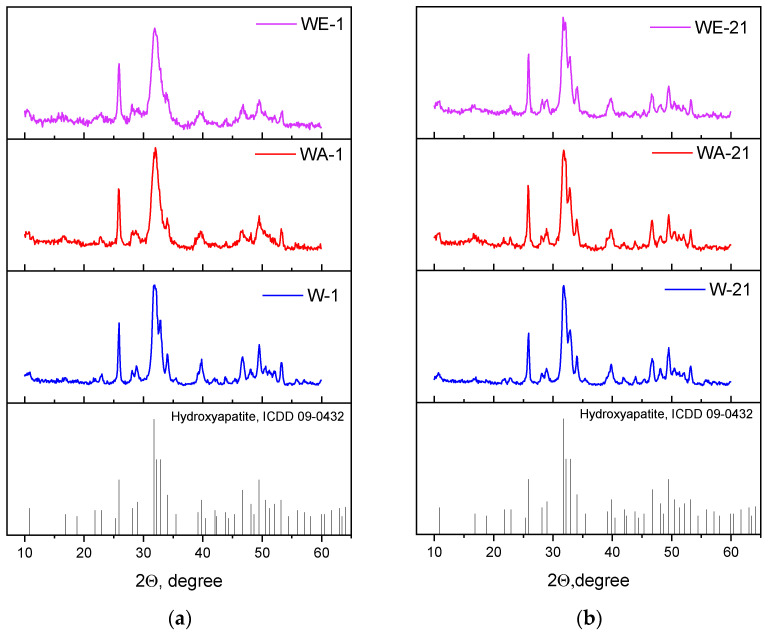
XRD data of the as-synthesized powders (**a**), and those data after 21 days of ripening (**b**).

**Figure 2 nanomaterials-13-00418-f002:**
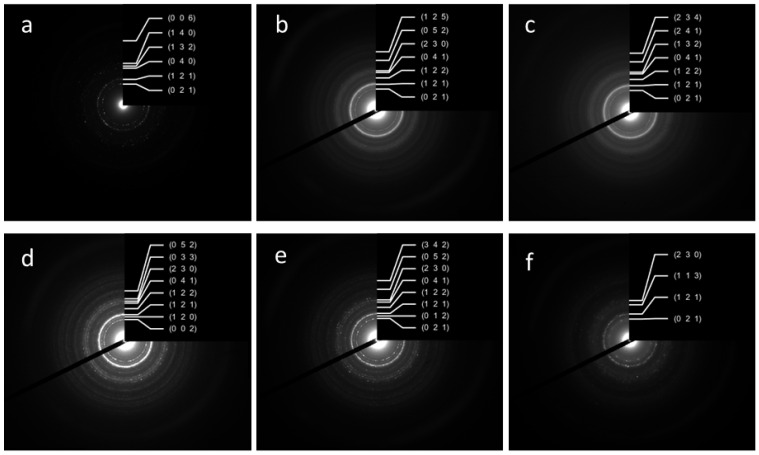
SAED ring patterns and Miller indexes of the powder samples W-1 (**a**), WA-1 (**b**), WE-1 (**c**), W-21 (**d**), WA-21 (**e**), WE-21 (**f**).

**Figure 3 nanomaterials-13-00418-f003:**
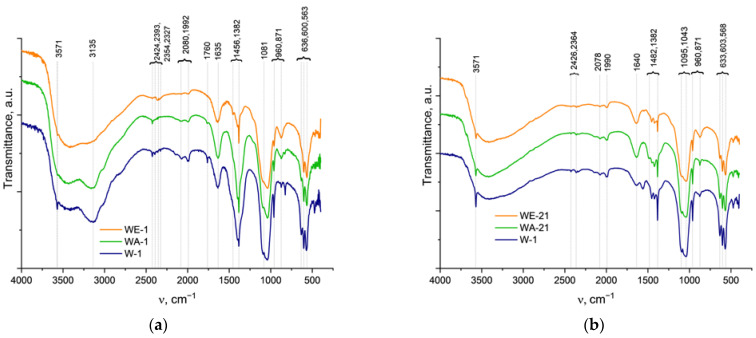
FTIR spectra of the as-synthesized powders (**a**), and those spectra after 21 days of ripening (**b**).

**Figure 4 nanomaterials-13-00418-f004:**
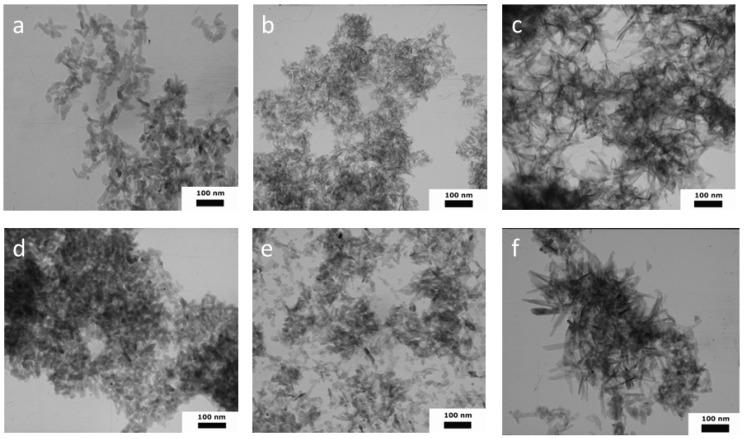
TEM images of HA powders synthesized in different media: W-1 (**a**), WA-1 (**b**), WE-1 (**c**), W-21 (**d**), WA-21 (**e**), WE-21 (**f**).

**Figure 5 nanomaterials-13-00418-f005:**
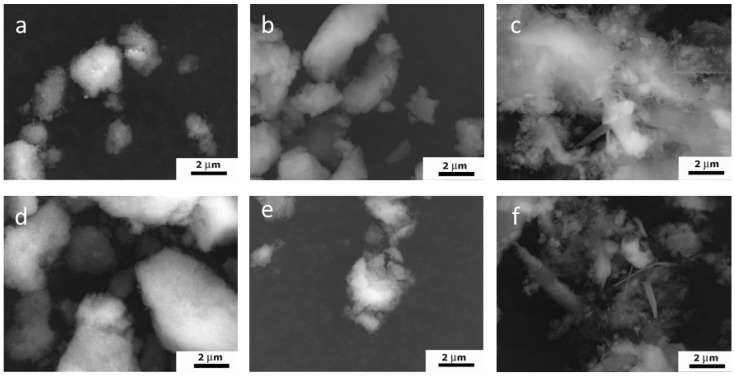
SEM images of HA powders synthesized in different media: W-1 (**a**), WA-1 (**b**), WE-1 (**c**), W-21 (**d**), WA-21 (**e**), WE-21 (**f**).

**Figure 6 nanomaterials-13-00418-f006:**
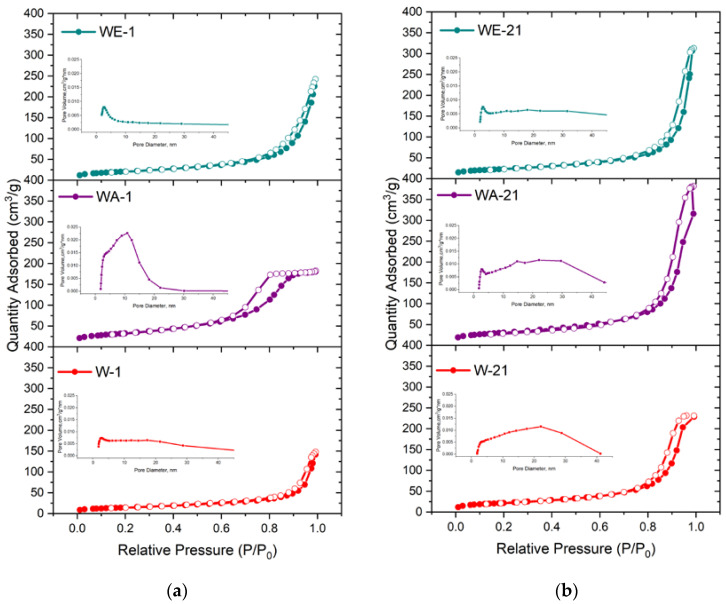
Nitrogen adsorption–desorption isotherms and BJH pore-size distributions of the as-synthesized powders (**a**) and those data after 21 days of ripening (**b**).

**Figure 7 nanomaterials-13-00418-f007:**
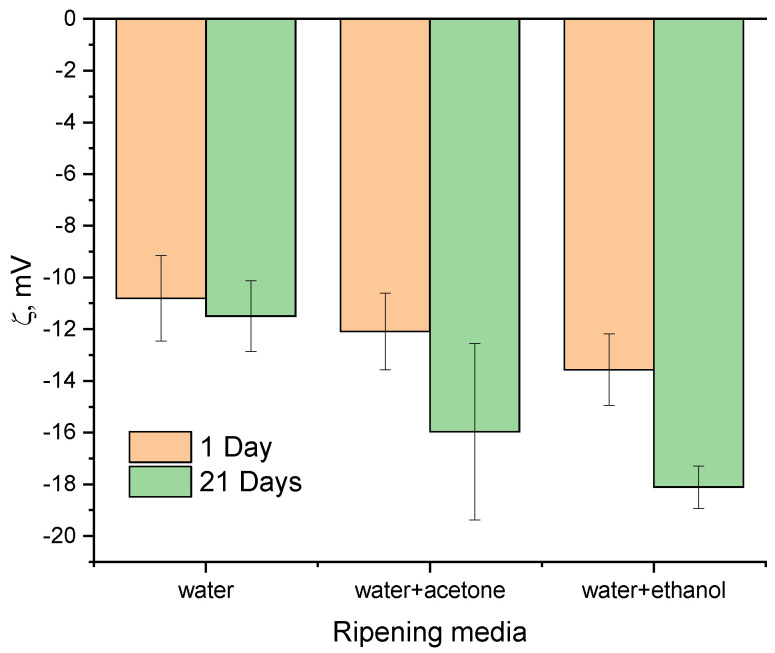
Zeta potential results of the as-synthesized powders and those data after 21 days of ripening.

**Figure 8 nanomaterials-13-00418-f008:**
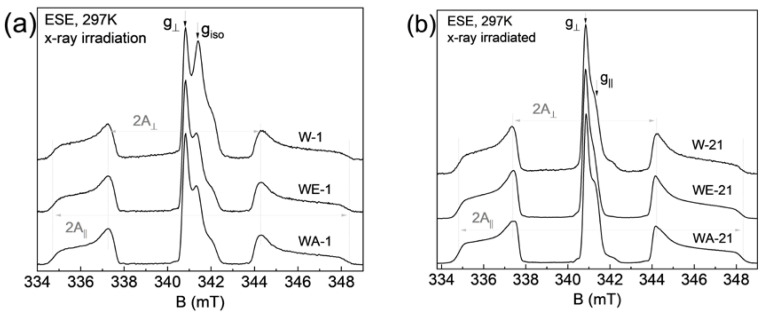
Room temperature, pulsed EPR spectrum of X-ray irradiated hydroxyapatite, where (**a**) a comparison of as-synthesized results for three different media is shown, (**b**) a similar representation of the results after 21 days of ripening, (**c**) a comparison of the WE (water-ethanol) results for the two ripening times, together with the corresponding theoretical simulation.

**Figure 9 nanomaterials-13-00418-f009:**
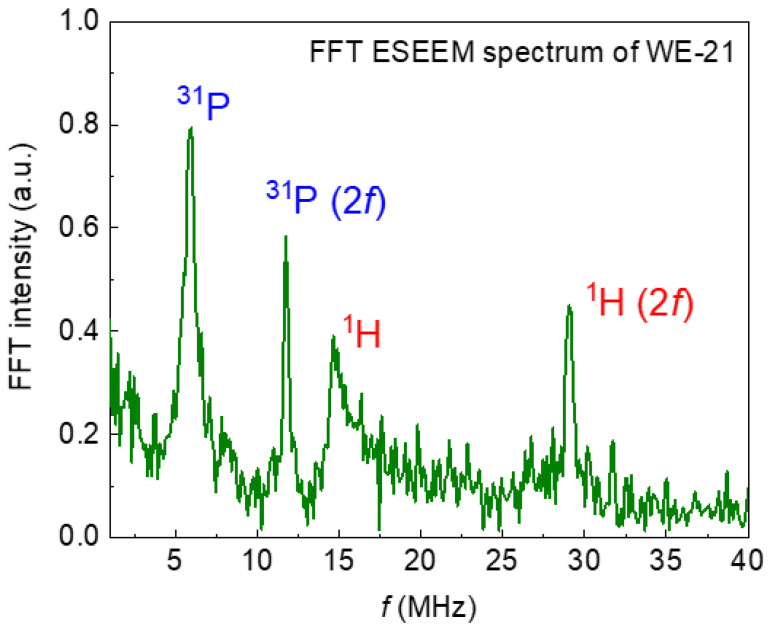
The spectrum of electron–nuclear interactions obtained by the ESEEM pulse sequence. Resonant absorptions in the frequency domain correspond to nearby magnetic nuclei that interact with the nitrogen radical in the crystal lattice of the sample. The (*2f*) indicates second-harmonic modulation.

**Table 1 nanomaterials-13-00418-t001:** Chemical compositions of the powders.

Sample Code	Ca^2+^ Concentration		P^3−^ Concentration		Ca/P
	mol/L	wt%	mol/L	wt%	
W-1	0.756	30.30	0.437	13.55	1.729
WA-1	0.850	34.07	0.482	14.94	1.763
WE-1	0.844	33.83	0.480	14.88	1.757
W-21	0.868	34.79	0.460	14.26	1.885
WA-21	0.899	36.03	0.477	14.79	1.884
WE-21	0.900	30.30	0.505	15.66	1.782

**Table 2 nanomaterials-13-00418-t002:** Characteristics of the lattice parameters and crystallite sizes of the powders.

Sample Code	a, nm	c, nm	c/a	V, nm^3^
Standard HA (ICDD #09-0432)	0.9418	0.6884	0.7309	0.5287
W-1	0.9439 (5)	0.6883 (3)	0.7292	0.5311
WA-1	0.9467 (7)	0.6881 (3)	0.7268	0.5341
WE-1	0.9447 (1)	0.6870 (2)	0.7272	0.5309
W-21	0.9434 (7)	0.6889 (3)	0.7302	0.5309
WA-21	0.9441 (6)	0.6889 (5)	0.7297	0.5318
WE-21	0.9444 (9)	0.6882 (5)	0.7287	0.5316

**Table 3 nanomaterials-13-00418-t003:** The textural parameters of the powder samples.

Sample	BET Specific Surface Area (m^2^g^−1^)	t-Plot Micropore Volume (cm^3^ g^−1^)	Total Pore Volume (cm^3^g^−1^)	Adsorption Average Pore Width (nm)
W-1	30.3	0.00364	0.080	10.6
WA-1	116.4	0.00302	0.286	7.9
WE-1	52.1	0.00125	0.231	16.5
W-21	79.4	0.00260	0.353	17.8
WA-21	109.7	0.00450	0.593	18.9
WE-21	83.2	0.00430	0.486	21.3

**Table 4 nanomaterials-13-00418-t004:** Spectroscopic parameters of the spin Hamiltonian (2) of the nitrogen radical in hydroxyapatite.

	$g_{⊥}$	$g_{∥}$	$A_{⊥}$	$A_{∥}$	ΔB Linewidth (mT)
1 day	2.0083	2.004	3.33	6.96	0.51
21 days	2.0083	2.004	3.25	6.67	0.31

## Data Availability

Data sharing not applicable.

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
