# Peer review of "Effects of Various Ripening Media on the Mesoporous Structure and Morphology of Hydroxyapatite Powders"

_nanomaterials, 2023, doi:10.3390/nano13030418_

Round 1

Reviewer 1 Report

The introduction section should be well structured, i.e., it should connect the importance of the study in a gradual manner: (1) provide background information and set the context, (2) introduce the specific topic of your research and explain why it is important, (3) mention past attempts to solve the research problem or to answer the research question and (4) conclude the introduction by mentioning the specific objectives of your research. Please apply.

Materials and Methods. More details regarding the elemental analysis by ICP-AES should be presented. Also, please uniformize/ describe all equipment used in the experiment – work development environment / work apparatus should be given – model of equipment (manufacturer, city, country).

Results. 3.1. Chemical composition – Please refer to % instead of mol/L.

Please improve the quality of Figures 6 and 8 and replace “,”by “.”when referring to numbers.

Results. The obtained results are difficult to follow. There is plenty of general information in this section. his issue should be corrected by highlighting the main findings, shortening the information presented,

Discussion. The comparison with previous studies is difficult to follow. There is plenty of general information in this section. This issue should be corrected by shortening the information presented, while keeping only the relevant data and compare the obtained results with similar (recent) studies.

Conclusions. In my opinion the conclusions are too general and unclear. Please revise.

Finally, I consider that the work is not suitable for publication in this form and requires large additions. If the manuscript will not be considerable improved, I will not recommend its publication. Nevertheless, the efforts of performing all the experiments have been significant and I hope that in the near future all the issues will be solved.

Author Response

We would like to thank the anonymous reviewer for comments on our manuscript.

The introduction section should be well structured, i.e., it should connect the importance of the study in a gradual manner: (1) provide background information and set the context, (2) introduce the specific topic of your research and explain why it is important, (3) mention past attempts to solve the research problem or to answer the research question and (4) conclude the introduction by mentioning the specific objectives of your research. Please apply.

Thank you for your comments. We improved the introduction section and added additional information. Shortly, we would like to focus out, that we described the increased importance of the development of mesoporous materials, including hydroxyapatite powders, and listed its application fields (1), we discussed the existence of expensive and comprehensive methods of the mesoporous HA synthesis and motivated the aim of the creation of the low-cost HA obtaining methods, (3) we described the attempts to obtain the HA with high textural properties (specific surface area and high porosity) by the template-based, organic modifier, and another synthesis routes as well as the influence of the introduction of the additional ions in the HA structure, (4) we described the specific of the ripening (maturation) approach which we planned to investigate and pointing out the applied characterization technique.

Materials and Methods. More details regarding the elemental analysis by ICP-AES should be presented. Also, please uniformize/ describe all equipment used in the experiment – work development environment / work apparatus should be given – model of equipment (manufacturer, city, country).

Thank you for your suggestion, additional explanation of ICP-AES experiment was provided and necessary information in the uniform for all the equipment applied was adjusted.

Please improve the quality of Figures 6 and 8 and replace “,”by “.”when referring to numbers.

Thank you, corrected.

Results. The obtained results are difficult to follow. There is plenty of general information in this section. his issue should be corrected by highlighting the main findings, shortening the information presented,

Discussion. The comparison with previous studies is difficult to follow. There is plenty of general information in this section. This issue should be corrected by shortening the information presented, while keeping only the relevant data and compare the obtained results with similar (recent) studies.

Concerning the complexity of the results and discussion section, we should state that we try to provide accurately step by step all the data obtained and the corresponding discussion with references to both newest and most competent works of leading researchers in the field. Despite the possible complexity of the above reasoning, it seems that the information presented and comparisons with close results are important for other researchers in this field, and we would like to keep these sections in the form in which they are presented now - with the appropriate level of detail and consideration of all important aspects of the experiment.

Conclusions. In my opinion the conclusions are too general and unclear. Please revise.

Thank you, we try to improve our conclusion.

Reviewer 2 Report

The specific surface area and porosity of mesoporous materials are important indicators and bases for carrying out adsorption and catalysis related applications.

The authors attempted to enhance the specific surface area and porosity of hydroxyapatite powders by changing the solvent medium and maturation time. By observing the changes in the performance of the samples after switching from water as the synthesis medium to water/ethanol and water/acetone as the synthesis medium with a maturation time of 21 days, it was found that the introduction of the additional solvent could enhance the specific surface area and the extension of the maturation time could enhance the porosity. The characterization carried out supports the observed phenomena.

The authors suggest that recrystallization of particles by decreasing the aspect ratio and the formation of smaller and more relaxed agglomerates caused by changes in the solid surface potential are essentially responsible for the change in specific surface area porosity.

Overall, the authors provide a very interesting solution for enhancing the porosity and specific surface area of hydroxyapatite and offer their own insight. I recommend its publication in the journal.

Author Response

We would like to thank the anonymous reviewer for considering our manuscript and for your good appreciation.

Round 2

Reviewer 1 Report

Most of the comments were either not fully addressed or at all.

Author Response

Most of the comments were either not fully addressed or at all.

We tried to improve our manuscript according to your recommendations.

The introduction section should be well structured, i.e., it should connect the importance of the study in a gradual manner: (1) provide background information and set the context, (2) introduce the specific topic of your research and explain why it is important, (3) mention past attempts to solve the research problem or to answer the research question and (4) conclude the introduction by mentioning the specific objectives of your research. Please apply.

Thank you for your comments. We one more time improved the introduction section and added additional information trying to focus on the importance of the new mesoporous materials creation. Also, we try to describe the aim of the investigations more clearly.

Materials and Methods. More details regarding the elemental analysis by ICP-AES should be presented. Also, please uniformize/ describe all equipment used in the experiment – work development environment / work apparatus should be given – model of equipment (manufacturer, city, country).

In the last version, an additional explanation of ICP-AES experiment was provided and necessary information was presented.

Please improve the quality of Figures 6 and 8 and replace “,”by “.”when referring to numbers.

Thank you, we corrected the figures in the last version. Thus, the symbols on the graphics were enlarged, and “,” were changed by “.”

Results. The obtained results are difficult to follow. There is plenty of general information in this section. his issue should be corrected by highlighting the main findings, shortening the information presented,

We tried to improve this section and shorten some discussion. We deleted Figure S2. At the same time, all our previous papers were written with the same may be too detailed results sections, and it is too difficult to change the storytelling style.

Discussion. The comparison with previous studies is difficult to follow. There is plenty of general information in this section. This issue should be corrected by shortening the information presented, while keeping only the relevant data and compare the obtained results with similar (recent) studies.

We tried to point out some of the most important statements to make the discussion clear.

Conclusions. In my opinion the conclusions are too general and unclear. Please revise.

Also, we rewrite the conclusion.

Best regards, Dr. Margarita Goldberg on behalf of other authors.

Round 3

Reviewer 1 Report

Authors have revised the manuscript accordingly, so, I would like to recommend that it is ready to be accepted.